# A Comparative Photographic Review on Higher Plants and Macro-Fungi: A Soil Restoration for Sustainable Production of Food and Energy

Hassan El-Ramady [1,2,*], Gréta Törős [2], Khandsuren Badgar [2,*], Xhensila Llanaj [2], Peter Hajdú [2], Mohammed E. El-Mahrouk [3], Neama Abdalla [4] and József Prokisch [2]

1   Soil and Water Department, Faculty of Agriculture, Kafrelsheikh University, Kafr El-Sheikh 33516, Egypt
2   Faculty of Agricultural and Food Sciences and Environmental Management, Institute of Animal Science, Biotechnology and Nature Conservation, University of Debrecen, 138 Böszörményi Street, 4032 Debrecen, Hungary; toros.greta@agr.unideb.hu (G.T.); xhenillanaj@gmail.com (X.L.); hajdu.peter@agr.unideb.hu (P.H.); jprokisch@agr.unideb.hu (J.P.)
3   Horticulture Department, Faculty of Agriculture, Kafrelsheikh University, Kafr El-Sheikh 33516, Egypt; threemelmahrouk@yahoo.com
4   Plant Biotechnology Department, Biotechnology Research Institute, National Research Centre, 33 El Buhouth St., Dokki, Giza 12622, Egypt; neama_ncr@yahoo.com
*   Correspondence: hassan.elramady@agr.kfs.edu.eg (H.E.-R.); b_khandsuren@muls.edu.mn (K.B.)

**Abstract:** The Kingdom of Plantae is considered the main source of human food, and includes several edible and medicinal plants, whereas mushrooms belong to the Kingdom of fungi. There are a lot of similar characteristics between mushrooms and higher plants, but there are also many differences among them, especially from the human health point of view. The absences of both chlorophyll content and the ability to form their own food are the main differences between mushrooms and higher plants. The main similar attributes found in both mushrooms and higher plants are represented in their nutritional and medicinal activities. The findings of this review have a number of practical implications. A lot of applications in different fields could be found also for both mushrooms and higher plants, especially in the bioenergy, biorefinery, soil restoration, and pharmaceutical fields, but this study is the first report on a comparative photographic review between them. An implication of the most important findings in this review is that both mushrooms and plants should be taken into account when integrated food and energy are needed. These findings will be of broad use to the scientific and biomedical communities. Further investigation and experimentation into the integration and production of food crops and mushrooms are strongly recommended under different environmental conditions, particularly climate change.

**Keywords:** phytoremediation; food crops; energy crops; polluted soils; plant mineral nutrients; phytomedicine



## 1. Introduction

A great challenge faces humanity in producing edible plants, which should contain enough amounts of mineral nutrients, and are required for human nutrition [1]. Although both plants and humans require, in general, the same mineral elements for their healthy growth and development, the ideal future crops for human nutrition should not include toxic elements in the edible parts [2,3]. Mineral elements in the soil are taken up by the plant roots and transported to the edible parts for human consumption through various different transporters. Therefore, several studies focus on the edible plants from different points of view for human health, including (*i*) studies of plant functional traits for human health, especially unconventional edible plants [4–6]; (*ii*) producing biofortified plants with a focus on the malnutrition/medicinal attributes [7–12], (*iii*) nutritional aspects of plant-based diets for human diseases [13–15], (*iv*) studies of plant secondary metabolites

and their extraction as bioactive compounds [16,17], (*v*) anti-nutrients of major plant-based foods [18], (*vi*) food security and plant nutrition under problems of climate change [19], and (*vii*) using mushrooms as bioindicators for pollution and its risks to health [20].

The production of enough food for the global population needs more efforts to exploit every inch to cultivate and produce foods, as well as energy at the same time, because food and energy are essential components for human life and sustainable development [21]. The production of food and energy gives rise to competition for cultivated soil. The arable land that is already used for the cultivation of foods should be increased for more food production, without any deducted lands for energy production. There is a difficult equation concerning the energy–food nexus, which should be solved as reported in many studies on the energy–food nexus from different points of view, such as the production of biofuel based on the water–food–energy nexus [22], rice production and its nexus of food–energy–emissions [23], the scarcity risk of the energy–food nexus [21], and sustainable dairy farming under the security of energy, food, and water [24]. New non-exploited areas such as polluted or marginal soils for energy production through soil restoration are considered sustainable solutions for producing energy [25].

Higher plants and macro-fungi (mushrooms) are important species, which have many common attributes (e.g., the nutritional and medicinal ones), although they have many differences. Higher plants can form their own food (which contains chlorophyll as autotrophic) from sunlight, water, and $CO_2$, whereas mushrooms as saprophytes can biodegrade dead organic matter by extracting enzymes [26]. Fungi are considered, in general, decomposers, pathogens, parasites, or mutualists [27].

Therefore, this review is a comparative study between mushrooms as macro-fungi and higher plants from mainly the human health point of view. This review includes also phytomedicine and its potential for human health, the unconventional foods derived from plants and mushrooms, soil degradation and its restoration by plants and mushrooms, the integrated production of food and energy, and finally the agro-integration between food crops and mushrooms.

## 2. Methodology of the Review

This review depended on collecting available published materials from the main websites of major publishers such as Frontiers, Elsevier (ScienceDirect), Springer, Multidisciplinary Digital Publishing Institute (MDPI) journals, medical publications at the U.S. National Institute of Health's National Library of Medicine (PubMed Central or PMC), etc. The main keywords were higher plants and macro-fungi, with the search broadened to include the words "sustainability", "plant nutrition", "unconventional food", "mushrooms", "soil restoration", "food crops", and "energy crops". The main steps used in this review included the building of a table of contents after intensive reading, searching suggested titles or sections, sorting the collected published materials (original articles, reviews, books, and chapters) based on the reputation of both authors and the journal, and then starting to write down the manuscript. At least five essential components of a successful review should be found; (1) the update and clear idea, (2) harmony and logical arrangement of their contents, (3) survey studies concerning some selected sections in the review through some comprehensive and very update tables, (4) photos and drawing figures are important in delivering the idea of the review faster than words, and (5) cited literature reviews, used in this review. This is the first report on a comparative photographic study on mushrooms and plants, which included many photos and drawing figures in this context.

## 3. Plant and Human Nutrition for Sustainability

Nutrition in plants and human presents many similarities and differences. It is the process of obtaining or providing the organism with food necessary for its growth and health. Several differences in the nutrition of plants and humans can be listed in Table 1. This nutrition has a strong link to Sustainable Development Goals as a part of the global public agenda, which can contribute to the structuring of global sciences and research [28]. Not surprisingly,

considering the lessons learned from Covid-19, the required scientific studies should focus on the area of sustainability and human health [28,29]. Richardson and Lovegrove [29] reported on the nutritional status of some micronutrients (Cu, Fe, Se, and Zn, and vitamin D, A, B vitamins, and vitamin C) and their possible and modifiable risk factor for COVID-19, which support the normal functions of the human immune system. They confirmed that avoiding deficiencies in the intakes of these micronutrients in patients could strengthen their resilience to the COVID-19 pandemic. Finally, a number of important nutrients need to be considered that could be found in vegetables, fruits, or edible plants (Figure 1).

**Table 1.** The main difference between the nutrition of both plants and humans.

| Comparison Item | Plant Nutrition | Human Nutrition |
|---|---|---|
| Forming own food | They can because of chlorophyll | They cannot |
| Main requirements for nutrition | Plants need sunlight, $CO_2$ and water as autotrophic | They metabolize large food molecules as heterotrophic |
| Final product from nutrition | Mainly glucose, energy, and oxygen | Amino acids, monosaccharides, fatty acids, and glycerol |
| Main organs involved in the nutrition | Leaves, including their components (chloroplasts, xylem, and phloem) | Mouth, esophagus, stomach, small intestine and the large intestine |
| Amino acids forming | Uptake of N, converted to $NH_3$, form amino acids and then proteins | Amino acids can be obtained from the breakdown of proteins |
| Getting energy | Both photosynthesis and respiration can be used for forming energy (ATP) | Only the respiration process can produce energy (ATP) |
| Storage of carbohydrates | Plants can store glucose in the form of starch | Humans can store glucose in the form of glycogen |
| Enzymes involved | During nutritional processes, many enzymes: amylase, cellulase, lipase, phosphatase, phytase and urease | During metabolism, the main nutrients are carbohydrates, forming glucose, lipids (fatty acid) & proteins (amino acids) |
| Essential elements for both plants/humans | C, H, O, N, P, K, Ca, Mg, S, Fe, Mn, Cu, Cl, Mo, Ni, Zn | C, H, O, N, P, K, Ca, Mg, S, Fe, Mn, Cu, Cl, Mo, Ni, Zn |
| Suggested as essential | Al, Co, Se, Si, Na | As, F, Si, V, Cr, Sn, Ni |
| Essential element (only) | B | Na, Co, I, Se |

Sources: [2,30–33].

Due to healthy nutrition being the backbone for human health, there is an urgent need to explain the advances in nutrition science during the next years under the "One Health" initiative [34]. Food supply chains should follow a sustainable food system (by reducing food wastes and losses) in vegetables, fruits, and mushrooms under a circular economy strategy [35–37]. The most important nutrition issues and their role in sustainability included the potential of functional food for human health, attaining sustainable food systems, the global crisis of production of healthy and sustainable proteins, nutrigenomics and personalized nutrition, immunometabolism and nutritional immunology, performance nutrition, managing old/new pandemics such as obesity and COVID-19, human gut microbiomes and nutrition, nutrition and brain functions, fasting and identifying alternative dietary strategies, human nutrition types (i.e., basic, applied, and clinical nutrition), and nutritional aspects of emerging technologies [18,28,38,39]. Therefore, several books have been published recently on the role of medicinal plants and human health, such as Frazier and Matthew [40], Goyal and Ayeleso [41], Suleria and Barrow [42], Goyal et al. [43], Goyal and Chauhan [44], Suleria et al. [45], Goyal et al. [46], and Masoodi and Rehman [14,15].

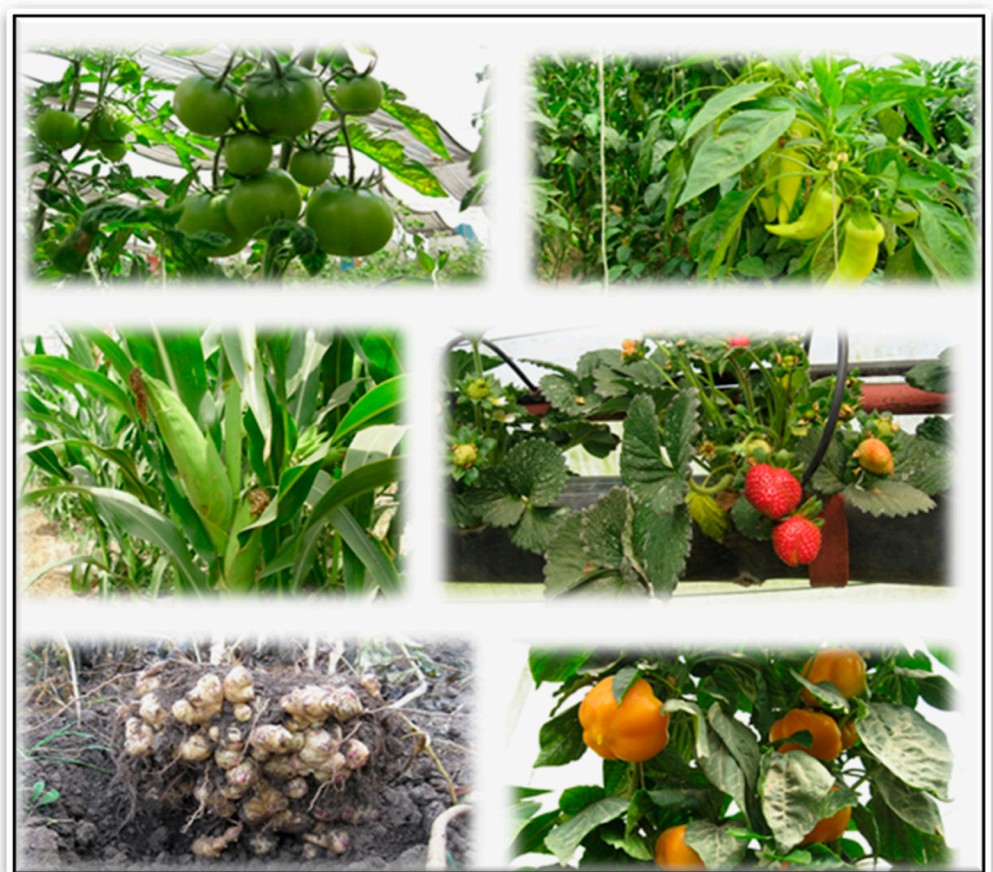

**Figure 1.** The problem of producing enough foods needs fertile soil, which supplies the cultivated plants with proper nutrients. Shown are photos of some edible plants, which can supply human with needed nutrients for human health. The photos in details from the upper photos (tomato and pepper), in the middle (maize and strawberry), and in the lower photos of Jerusalem artichoke tubers and fruits of color pepper, which show a good nutritional status as an important source for human health. All photos by **El-Ramady**.

As an important component of the human diet, proteins derived from plants are considered more sustainable sources compared to protein-derived from animals, because plant proteins have many eco-benefits including higher eco-sustainability to maintain eco-stability, greater food safety, fulfilling higher consumer needs, food affordability, and combating of "protein–energy malnutrition" [47]. Several studies have been published on plant-based proteins as a good source of many essential amino acids (serine, glycine, valine, alanine, cysteine, etc.), vital macro-nutrients, and are sufficient to achieve complete protein nutrition to sustain a better life for humans [19,48–51]. Along with providing amino acids in foods, protein can play a potential role in food formulations due to their distinguishing properties, including emulsification, water holding, gelling, foaming, thickening ability, and fat absorption capacity [52,53]. The main sources of plant-based proteins include many crops such as cereals (wheat (9.3–12.3%), rice (5.8–11%), maize (9–11%), barley (12%), and sorghum (11%)), legumes (chickpea (19–27%) pea (23–31%), soybean (37–44%), kidney bean (22–32%), faba bean (31%), lentil (23–36%), lupin (32–55%), and cowpea (28%)), pseudo-cereals (amaranth (14.5%), buckwheat (14.8%), and quinoa (13%)), nuts (peanuts (25–29%), almonds (29.9%), cashew nut (22.7%), Brazil nuts (19.7%)), and seeds (cottonseeds (38–45%), flax seeds (21–26%), sunflower (21–32%), pumpkin seeds (36.5%), and sesame seeds (18%)) [19,54,55]. Besides seaweed as a sustainable aquatic plant-based protein [3], and the previous non-traditional sources, there are many alternative plant-based proteins as reported in many publications [56–59]. These alternatives may include pea protein

isolate (86%) [60], pulse protein ingredients [55], and using proteins of lentil and quinoa alternative proteins in dairy [61]. There is also an ongoing need for global abundant sources of high-quality proteins for human nutrition. The plant's nutritional quality is controlled by several factors as presented in Figure 2. These factors include soil, plant climate, and environmental conditions. These factors may also control the kind of protein in many crops such as cereal proteins, which are low in their contents of amino acids (e.g., lysine, tryptophan, and threonine), whereas the proteins of vegetables and legumes have a lower amount of S-containing amino acids such as cysteine and methionine [47].

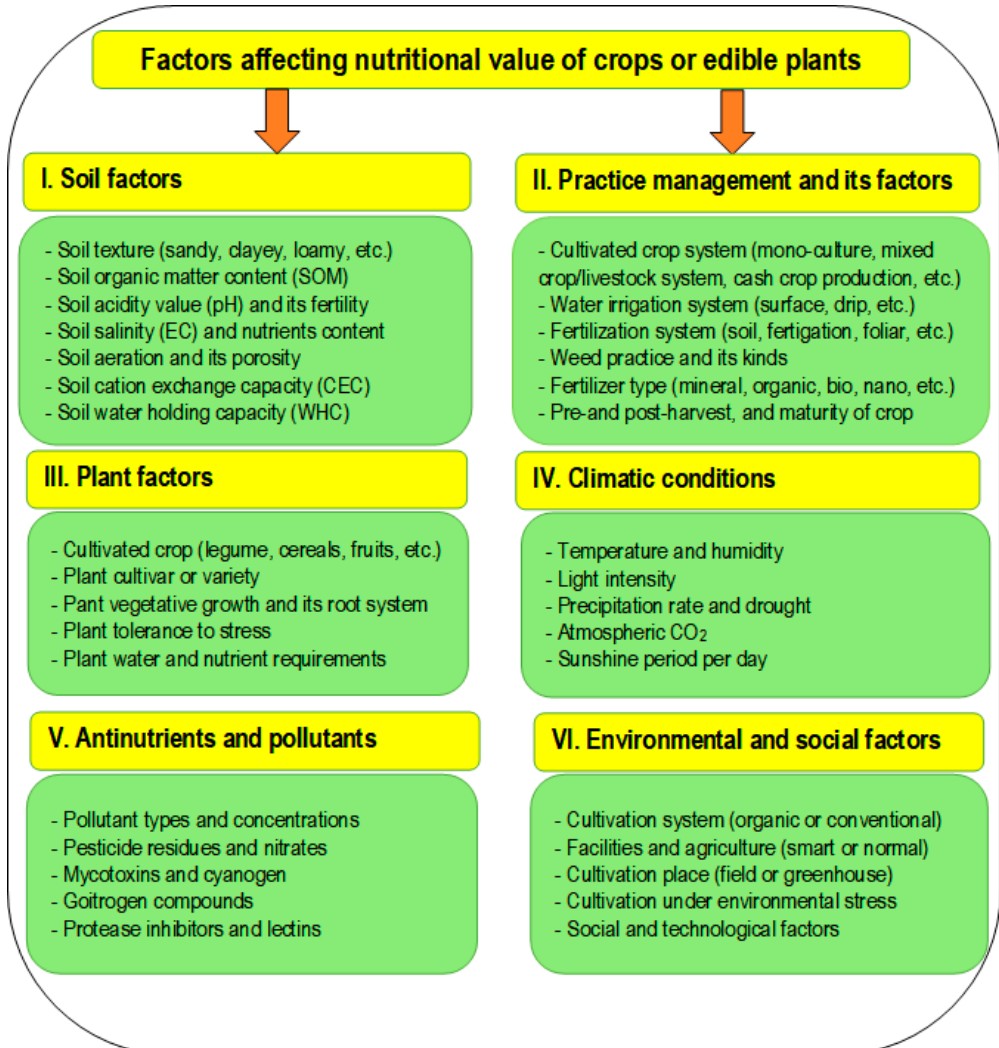

**Figure 2.** Factors affecting crop nutritional quality include factors related to cultivated crops, climate, and environmental and agricultural practices. More summarized information about antinutrients and pollutants is listed as well.

## 4. Phytomedicine and Human Health

The science of producing medicines from herbs or medicinal plants could be defined as phytomedicine. The historical background of this science may date back to early in human evolution or several thousand years ago, when humans isolated, extracted and purified many drugs from medicinal plants in ancient times for human health [62]. Phytomedicine also involves all clinical, pharmacokinetic, pharmacological, and toxicity-based studies of medicinal plants besides the exploring of different mechanisms of herb extracts [63]. Medicinal plants are in a continuous process of exploration, resulting in the unearthing of novel plant-based pharmaceuticals to understand the molecular mechanisms of conventional medication and its active ingredients, which may help the renovation of

plant-derived medications and detection of novel phyto-agents [63]. Phytomedicine may include several issues such as using medicinal plants as wound healing agents [64,65], natural anti-microbials from plants [66], herbal therapy or remedies [67], herbal cosmeticology [68], phytopharmaceuticals [69,70], phyto-pharmacology [71] using the herbal bioactivities in drug delivery systems like the ocular [72], pulmonary [73], transdermal [74], and vaginal and rectal drug delivery systems [75]. Therefore, it is very important to know exactly what we eat to stay healthy for a long time as conveyed by the term "nutraceuticals" [76]. This term was coined by Stephen De-Felice, who referred to pharmaceuticals and nutrients. There are many different kinds of nutraceuticals, such as dietary supplements, functional foods, dietary fibers, medical foods, prebiotics, and probiotics. Nutraceuticals can improve human health through enhancing the absorption of nutrients, supporting the micro-flora of the gastrointestinal system, and increasing detoxification. Nutraceuticals may have some limitations, including their slow mode of action and lack of strict control over the quality and concentration of ingredients [76].

Chinese herbal medicines are very common and widely used for treating several diseases in Chinese people as a source of bioactive ingredients, which have been extracted from herbs for therapeutic properties. The most well-known example of this medicine is the use of artemisinin to inhibit malaria by Nobel laureate Youyou Tu in 2015 [77]. Many medicinal plants can produce essential oil, which is considered a rich product of bioactives such as royal jasmine (*Jasminum grandiflorum* L.) and orange trees (*Citrus aurantium* L.) (Figure 3). Several bioactive ingredients have already been extracted from many natural medicinal plants such as ailanthone, artesunate, berberine, baicalin, curcumin, corylin, oridonin, triptolide/triptonide, shikonin, paeoniflorin, paeoniflorin, and soybean isoflavones (Figure 4). Bioactive compounds and pharmacological activities in some medicinal plants differ from plant to plant as presented in Table 2. More common medicinal plants are presented in Figure 5, including the basil plant (*Ocimum basilicum* L), scotch marigold (*Calendula officinalis* L.), rose geranium (*Pelargonium graveolens* L.), damask rose (*Rosa damascena* Mill.), black cumin (*Nigella sativa* L.), moringa (*Moringa oleifera* Lam.), mangrove (*Avicennia marina* (Forssk.) Vierh.), and jojoba (*Simmondsia chinensis* (Link) C. K.).

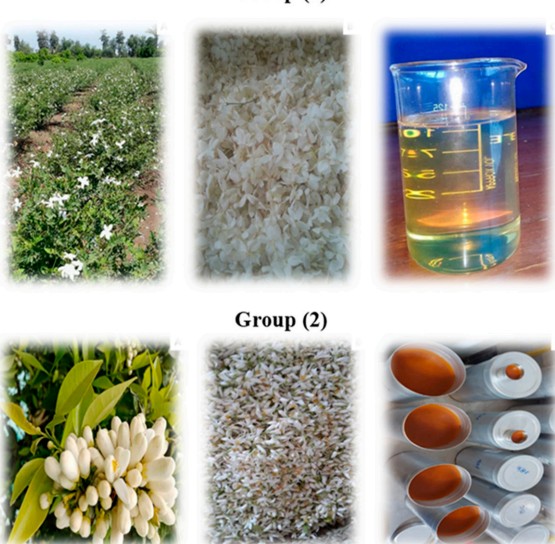

**Figure 3.** Medicinal plants are an important source for human health. Photos from left to right in **group (1)** represent the oil extraction from royal jasmine *(Jasminum grandiflorum* L.) flowering plants, whereas the second and third photos represent harvested flowers and jasmine oil. Photos in **group (2)** from left to right: oil extraction from bitter orange trees (*Citrus aurantium* L.) as flowering plants in the first photo, whereas the harvested flowers and Neroli oil can be noticed in the second and third photos, respectively. All photos by **El-Mahrouk**.

**Plant bioactives and their applications**

**Main plant bioactive groups**

The pharmacologically active components
that could isolate from plants, such as
- Flavonoids
- Carotenoids
- Glucosinolates
- Phenolic compounds
- Tannins
- Terpenoids

**Bioactive ingredients in Chinese herbal medicines**

- Ailanthone
- Berberine
- Triptolide/ Triptonide
- Shikonin
- Paeoniflorin
- Soybean Isoflavones
- Artesunate
- Baicalin
- Curcumin
- Corylin
- Oridonin
- Paeoniflorin

**Development of bioactive delivery systems**

- Therapeutic updates and future prospects
- By pressurized fluids-modern techniques
- Using modern extraction techniques
- Using a metabolomics approach
- Using green analytical chemistry
- Standardization of herbal bioactives
- Regulatory considerations of formulations

**Main applications of phyto-bioactives**

- Herbal bioactive–based cosmetics
- Nano drug delivery systems
- Wound healing applications
- Neurodegenerative disorders
- Gastrointestinal disorders
- Pulmonary/ocular drug delivery systems
- Transdermal drug delivery systems
- Vaginal and rectal drug delivery systems

**Main therapeutic of bioactive ingredients**

They have potential as treatment agents in
diseases such as
- Cancer, cardiovascular disease,
- Nervous system disease,
- Inflammatory bowel disease, and
- Infectious diseases

**Drug delivery of plant herbal bioactives**

- Nanoemulsions
- Microemulsions
- Liposomes / Niosomes/ Phytosomes
- Solid dispersions
- Nanogels/hydrogel
- Solid lipid nanoparticles
- Other NDDS herbal formulations

**Figure 4.** Medicinal plants have many bioactive compounds such as phenols, carotenoids, etc. These phyto-bioactives have many applications for human health. Due to bioactive ingredients in herbs, the phytomedicine is well known, especially in Chinese medicine which is famous all over the world, very common, and in general a lot of technologies in drugs or pharmaceuticals could be developed every day particularly using the nanotechnological approaches. "Chinese herbal medicines were the main treatment method used in ancient times by the Chinese to combat disease. As early as the Qin and Han Dynasty (around 221 BCE to 220 CE), Sheng Nong's Herbal Classic recorded 365 medicines. By the time of the Ming Dynasty (1368–1644), the number of CHMs listed in the book of Compendium of Materia Medica had increased to 1892" as reported by Dong et al. [77].

**Table 2.** List of bioactive compounds and pharmacological activities in some medicinal plants.

| Plant Species (Family) | Bioactive Phytochemicals | Pharmacological Activities | Refs. |
|---|---|---|---|
| Ginger: *Zingiber officinale* (Zingiberaceae) | Phenolic compounds (gingerols, paradols, and shogaols), flavonoids, carbohydrates, proteins, and terpenes | Antiemetic, anti-inflammatory, antidiabetic, anticancer, cardio-protective, and neuroprotective | [78] |
| Chewing stick or miswak: *Salvadora persica* Linn. (Salvadoraceae) | Underground parts or roots are used as toothbrushes, due to tannins (tannic acid), alkaloids (salvadorine), essential volatile oils, Vitamin C | Antimicrobial, antidiuretic, tick-repellent, anticancer, anti-inflammatory, hypolipidemic, and analgesic activities | [6] |
| Black seed: *Nigella sativa* (Ranunculaceae) | Seeds contain fixed oil (arachidonic, linoleic), protein, alkaloids, volatile oil (anethole, cymene), and saponin | Hepatoprotective, anti-cancer, anti-nephrotoxic, anti-diabetic, anti-parasitic, anti-malarial, anti-inflammatory and analgesic, etc. | [79] |
| Saffron: *Crocus sativus* L. (Iridaceae) | Flavonoids (flavone, flavonone), di-, mono-, tri-, tetra-terpenes (lycopene, crocetin) phenolics, carboxylic acids, phytosterols, vitamins (riboflavin) | Antiparasitic, antibacterial, hypotensive, antidepressant, anxiolytic, anticonvulsant, anti-Alzheimer, antitumor, anti-nociceptive, cytotoxic activity | [80] |
| Chicory: *Cichorium intybus*, (Asteraceae) | Vitamins (ascorbic acid, thiamine, riboflavin, retinol), carotenoids, inulin, niacin, sesquiterpenes, esculin, esculetin, cichorin A, lactucin, and lactucopicrin | Anti-inflammatory, antidiabetic, antimicrobial, gastroprotective, antioxidant, antimalarial, anthelmintic, analgesic and hepatoprotective activity | [81] |
| Basil plant: *Ocimum basilicum* L. (Lamiaceae) | Phenolic acids, isoprenoids, and flavonoids | Antioxidant, antibacterial, antifungal and anti-inflammatory activity | [82,83] |
| Scotch marigold: *Calendula officinalis* L. (Asteraceae) | Triterpenoid, carotenoids, lutein auroxanthin, zeaxanthin, saponins, beta-carotene, flavonol glycosides | Anti- genotoxic, anti-viral, and anti-inflammatory properties | [84] |
| Damask rose: *Rosa damascena* Mill. (Rosaceae) | Essential oil has β-citronellol, citronellol, docosane, geraniol, heneicosane, and nonadecane, | Rose oil is antiviral, anti-cancer, antioxidant, laxative, antiseptic and anti-inflammatory | [85] |
| Moringa: *Moringa oleifera* Lam. (Moringaceae) | Flavonoids, alkaloids, phenolics, tannins, saponins, glucosinolates vitamin A, vitamin C, Ca, and K | Anti-cancerous, cardiovascular, anti-asthmatic, antidiabetic, anti-microbial and anti-inflammatory | [86,87] |
| Mangrove: *Avicennia marina* (Forssk.) Vierh. (Acanthaceae) | Polyphenols, tannins, eicosanoic acid, cis9-hexadecenal, oleic acid, and di-Ndecylsulfone | Anti- antiviral, antibacterial, antifungal and antioxidant activities | [88,89] |
| Jojoba: *Simmondsia chinensis* (Link) C. K. (Simmondsiaceae) | Phenolic compounds like gallic acid, flavonoid, stigmast-5-en-3-ol, cis-9-octadecen-1-ol, 9-octadecen-1-ol, (Z), ergost-5-en-3-ol, (3-β)-ol, (Z)-14-tricosenyl formate, | Antidiabetic, anti-inflammatory, anthelminthic, antirheumatic, antiepileptic, antpsoriatic, antigonorrheal, analgesic, and pesticidal activities | [90,91] |
| Hibiscus: *Hibiscus asper* (Malvaceae) | Alkaloids, flavonoids, glycosides, phenols, saponins, steroid, tannin, terpenoids, 9, 12, 15-Octadecatrien-1-ol (Z, Z, Z) | Antiapoptotic Neuroprotective Antibacterial, anti-inflammatory, anti-ulcer, and anti-oxidative properties | [87,92] |

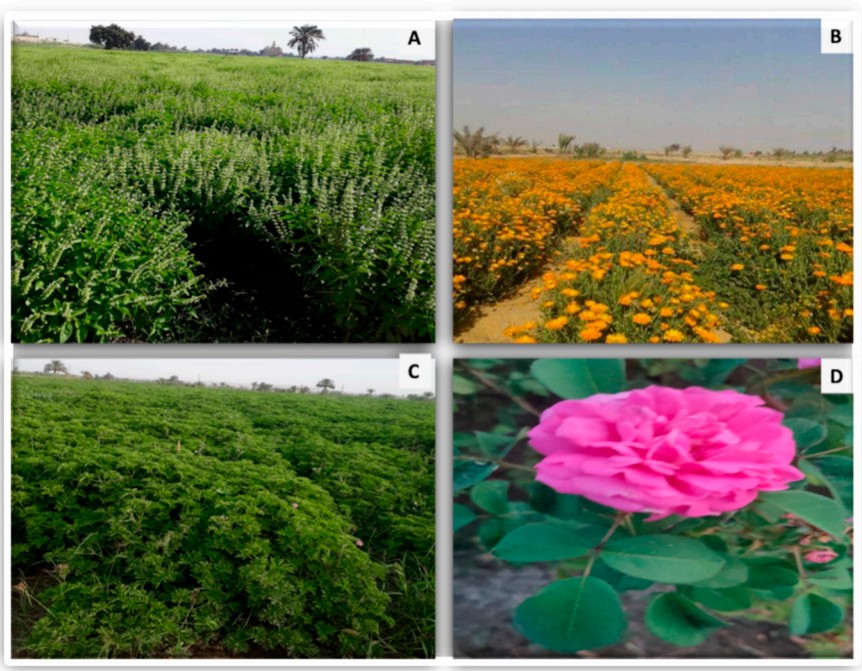

**Group (1)**

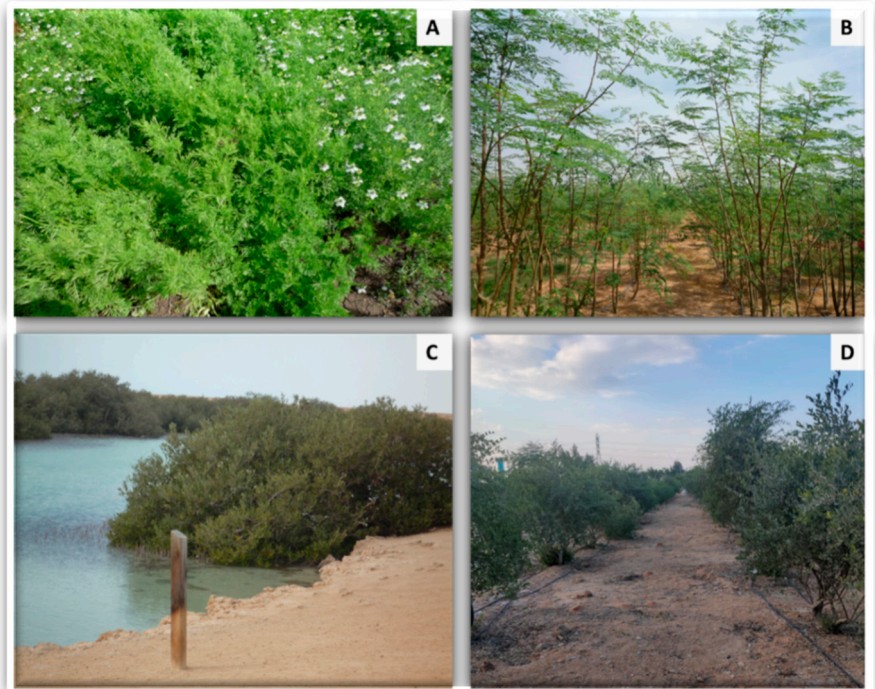

**Group (2)**

**Figure 5.** Medicinal plants are important source for human health as shown in this group of photos. Photos in **group (1)** include from up left to right (**A**) basil plant (*Ocimum basilicum* L), (**B**) scotch marigold (*Calendula officinalis* L.), and down left to the right (**C**) rose geranium (*Pelargonium graveolens* L.), and (**D**) damask rose (*Rosa damascena* Mill.). Photos in **group (2)** represent some medicinal plants including (**A**) black cumin (*Nigella sativa* L.), (**B**) moringa (*Moringa oleifera* Lam.), (**C**) mangrove (*Avicennia marina* (Forssk.) Vierh.), and (**D**) jojoba (*Simmondsia chinensis* (Link) C. K.). All photos by **El-Mahrouk**.

Many challenges face the regulation of herbal bioactive-based formulations, including challenges of safety, pharmacovigilance, quality, standardization, and clinical trials, as well as a lack of knowledge about herbal medicines [93]. In Table 2, the most common bioactive compounds in some medicinal plants and their pharmacological activities could be noticed. These bioactives may include main groups (i.e., phenolic compounds, flavonoids, carbohydrates, proteins, and terpenes), which have distinguished pharmacological activities such as antimicrobial, antidiuretic, anticancer, anti-inflammatory, analgesic and hypolipidemic activities. The main differences among these medicinal plants may be represented in the common kind of bioactives, such as phenolic compounds (i.e., gingerols, paradols, and shogaols) in ginger, tannic acid and alkaloids (e.g., salvadorine) in miswak, and others (Table 2).

It is well known that plants have been consumed by humans for thousands of years as a source of daily necessities and food supply. Plants also have been applied in several fields, including infrastructure, papermaking, production of perfumes and spices as well as applications for the treatment and prophylaxis of various diseases [94]. Therefore, traditional herbal medicine can be expressed as the use of plants or herbs as remedies in medicine in the following forms of herbal materials, herbs, herbal preparations, and finished herbal products containing active ingredients derived from plants/plant materials, according to the definition of the WHO [94]. Nano-formulated herbal bioactive could be applied for treating many human diseases such as neurodegenerative diseases, which can be nano-formulated using curcumin, quercetin, resveratrol, rutin, piperine, gallic acid, ferulic acid, and selenium [95]. There are many systems for delivering herbal bioactive-based nano-drugs, such as liposomes, nanoemulsions, niosomes, phytosome, polymeric micelles, nanoparticles, nanogels/hydrogel, and other novel drug delivery systems herbal formulations [87].

## 5. Higher Plants and Mushrooms: A General Comparison

Higher plants have many similarities and differences to macro-fungi (mushrooms) as presented in Figure 6A,B. Both higher plants and mushrooms are living organisms belonging to one domain (Eukarya) and are considered, in general, vegetarian as well as both of them possessing distinguishing attributes for human health. High plants are located in the Kingdom of Plantae, but mushrooms are in the Kingdom of Fungi. They can also be used as edible sources for medicinal activities.

There are also many differences between them, especially the nutrition mode, which depends on their content of pigments or chlorophyll, as well as the reproduction method, and the main structure of each one. From this point, mushrooms are fungi organisms that have no chlorophyll; thus, they cannot form their own food, but they are saprophytes (can release some enzymes to biodegrade organic matters and convert them into simple compounds to obtain their necessary foods). The main method for reproduction of mushrooms is by spores, and not all mushrooms can be cultivated like plants [26,96,97]. Some species of both higher plants and mushrooms have nutritional and medicinal attributes, and are called medicinal plants or mushrooms. There are several kinds of mushrooms, which can in general be categorized into edible, medicinal and poisonous mushrooms, as reported by El-Ramady et al. [36]. More dimensions for the sustainable applications of mushrooms could be found in Elsakhawy et al. [98] and El-Ramady et al. [99] whereas the sustainable production of medicinal plants is a great challenge, especially under the adverse conditions reported in detail by Aftab [100].

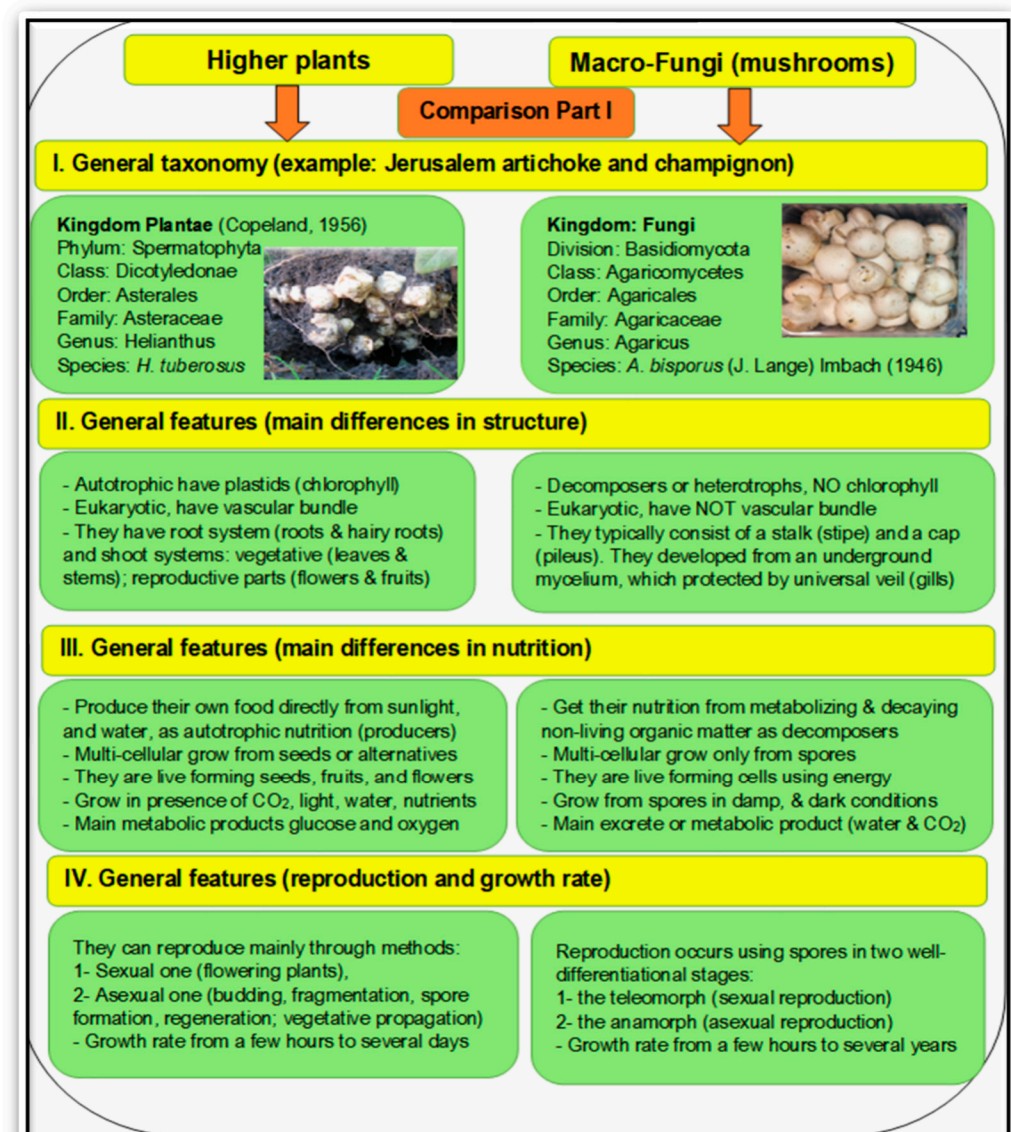

(**A**)

**Figure 6.** *Cont.*

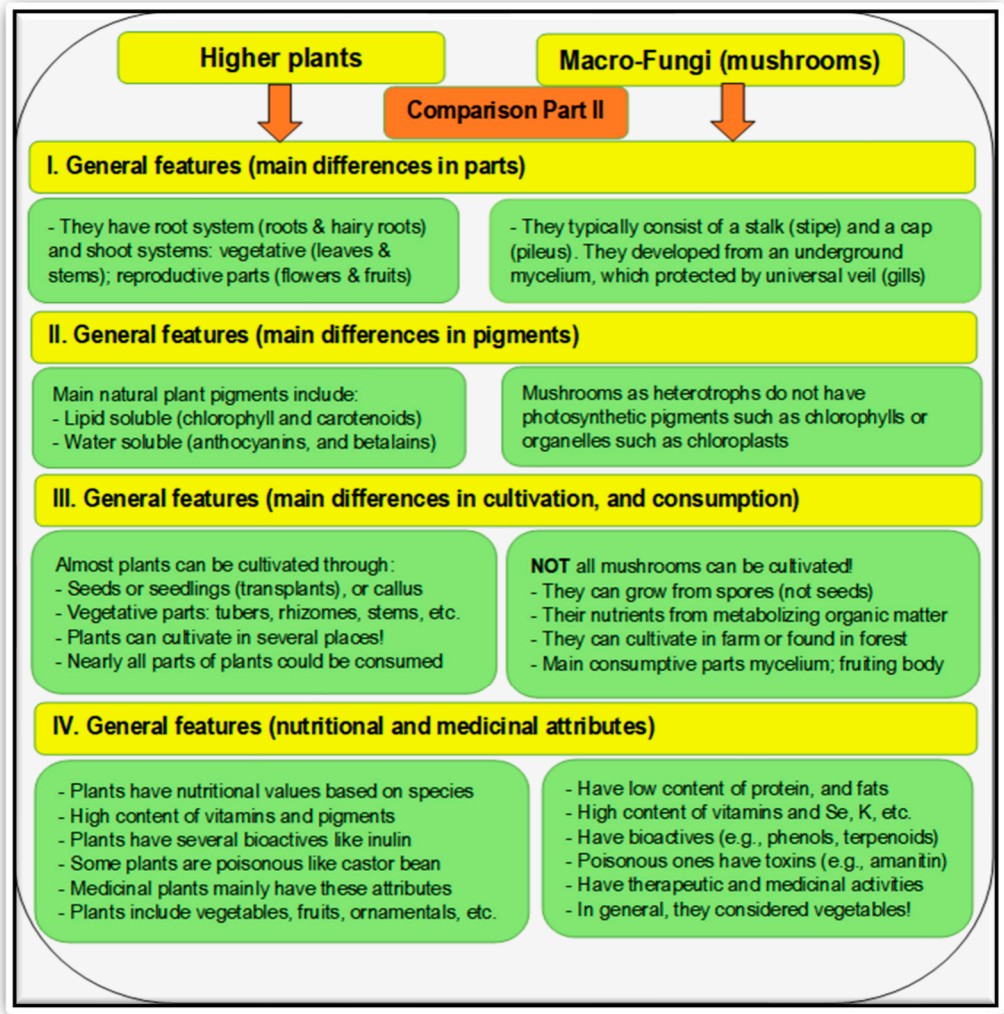

**(B)**

**Figure 6.** (**A**) This is the first part of a comparison between higher plants and mushrooms from different points of view, such as taxonomy (using Jerusalem artichoke as an example of a higher plant and champignon as a mushroom), structure, nutrition, reproduction and growth. (**B**) This is the second part of a comparison between higher plants and mushrooms including more different points of view such as the pigments, cultivation and their nutritional/medicinal attributes.

## 6. Unconventional Foods of Plants and Mushrooms

There are several classifications of plants, which depend on a specific categories such as higher and lower plants based on the existence of flowers or vascular system; common and unconventional edible plants; traditional and modern wild edible plants [101]; cultivated and wild edible plants [37], and conventional and unconventional food plants [102]. The common attributes that can be existed in edible mushrooms and edible plants are being consumable foods and having a nutritional value, such as desirable content of proteins, carbohydrates, or vitamins. Many studies on unconventional foods derived from plants or mushrooms have been published in different locations all over the world for plants [102–104] or mushrooms [105,106]. Increasing attention has been noticed to this group of underutilized plants, which have many different terms such as unconventional vegetables or traditional vegetables, alternative food plants, famine foods, wild edible plants, and plants for the future [103,107]. Therefore, there is an urgent need for unconven-

tional medicinal/food plants with consideration of their potential under the initiative of "from flask to patient" and "from field to fork" [108].

There are many innovative technologies in food production acting as food frontiers, which can achieve eco-sustainability and the security of global food, seeking for more sustainable future. These food frontiers may include controlled-environment agriculture [109], climate-driven northern agricultural expansion [110], cellular agriculture [111], entomophagy [112] and seaweed aquaculture [3,113,114]. Based on the single-cell protein in macro-fungi/mushrooms, many possibilities exist to use agricultural residues and wastes because of their fast growth, high cell densities, long history of use, and simple reactor design. However, they have many challenges, including a need for non-food carbon substrates and a possibility for existing mycotoxins [111]. The main target that attracts several scientists all over the world is how to find "plant protein-based meat and dairy analogues" especially under climate change [115–117], and single-cell proteins derived from mushrooms as reported by Stephan et al. [118] in Table 3. Mushrooms contain many bioactive compounds (e.g., phenolics, polysaccharides, polyketides, steroids, triterpenoids, etc.), are considered nutraceuticals, a vegan protein source (up to 45%), and food flavor agents for the food industry [119]. The protein content in mushrooms depends mainly on the mushroom species and the edible part of the mushroom (i.e., fruiting body and mycelium), where the most common mushrooms *Pleurotus ostreatus, Agaricus biosporus*, and *Lentinus edodes* (Berg) have protein (%) in fruiting body and mycelium as follows (36 and 25.70), (45.9 and 47.1), and (23.5 and 17%), respectively in fruiting body and mycelium [119].

**Table 3.** Some examples of plant protein-based meat and dairy analogues and their sources.

| Source of Proteins (the Country, if Any) | Sources of Plant Proteins | Refs. |
|---|---|---|
| Dairy-based protein alternatives (general study) | Quinoa and lentil are considered high-digestibility proteins | [61] |
| Meat analogues including steak, burgers, meatballs, and cutlets (Italy) | Plant steaks, burgers, meatballs, and cutlets | [120] |
| Fibrous meat analogues (Poland) | Pea protein isolate and oat fiber concentrate | [121] |
| Innovative approaches for meat production | Strategy of adding quinoa or chia to meat products | [122] |
| Dairy cheese analogs (the USA) | Plant-based cheese analogs | [123] |
| Fermented meat sausages (Span and Italy) | Plant-based alternatives includes flavor of plant protein isolates | [124] |
| Applied binders in meat product (sausages) processing | Quinoa flour could be applied as binder in beef sausage production | [125] |
| Producing beef burgers formed from flour of quinoa and buckwheat | Flour of both quinoa and buckwheat along with soy protein in beef burgers | [126] |
| Meat co-products as a meat replacer (general study) | Crops or seaweeds can be replaced by 20% in meat protein | [127] |
| Boiled meat sausages (Germany) | *Pleurotus sapidus* as protein in a vegan boiled sausage analog | [118] |

On the other hand, many plants are considered plant-based foods that are rich in their content of proteins such as legumes, grains (mainly quinoa), nuts, and certain fruits like apricots, avocados, guavas, peaches, and raspberries [116]. Quinoa, as a pseudo-cereal crop, is considered important protein crop because of its amino acidic profile, gluten-free, high antioxidant content, bioactive properties, high nutrient content (i.e., Ca, P, B, Fe, K, Mg) and vitamins like B1, B2, B3, B6, and E [116]. Quinoa can also be used as an alternative for vegan diets, including in quinoa-based gels [128,129], quinoa protein isolates [130,131], to produce high-quality protein and low-cost enriched pasta [132], and quinoa protein fortification [133].

## 7. Soil Restoration by Plants and Mushrooms

It is well known that soil is one of the essential components of all life, and our life mainly depends upon it. Soil, as a non-renewable resource, can sustain our life on the Earth by supporting about 95% of global food production [134]. Soils have some essential ecosystem services, such as filtration of pollutants, purification of water, biomass production, and transfer of energy and mass between spheres [134]. In his wonderful book "*The Soil–Human Health Nexus*", Lal [135] stated that human health is indivisible, and has a very strong link to the health of the soil, plants, animals, and the entire ecosystem,

based on the initiative of "One Health". He added this ancient proverb "*When food is right, medicine is of no need; when food is wrong, medicine is of no use*". Therefore, there is a strong need and a global consensus to implement a "*soil protection policy*", enact the "Soil quality act" and respect the "Rights of soil" [135]. Due to misuse of soil resources and soil mismanagement, soil degradation has become a global issue, which impacts the entire environment or ecosystems and human health as well. Thus, the soil–human health nexus is considered a very important global issue that can never be over-emphasized.

Soil is usually formed under different soil forming factors (i.e., time, topography, organisms, climate and parent material), and forming processes. Based on the kind of parent material, the type of formed soil is determined to form sandy, loamy, clayey, calcareous soil, etc. (Figure 7). Several soils suffer from deterioration, which leads to degraded soil and results from many human activities such as urbanization, industrialization and civilization (Figure 8). Degraded soils are generally common under different conditions, which are represented in the decline of soil organic matter, soil acidification and its compaction, the toxicity of pollutants such as heavy metals, intensive use of chemical fertilizers and pesticides, low vegetation, and nutrient deficiency (Figure 9). These conditions may change the dynamics of plant–soil interactions and their outcomes.

According to FAO [136], the term land degradation is preferable compared to soil degradation and soil erosion, because it has a wider scope, covers all negative changes in the ecosystem capacity, and provides different services and goods from biological, social and economic aspects. In this section, soil degradation and its restoration by plants and mushrooms (Figure 10) are discussed. There are many human activities that lead to soil degradation all over the world; for example, in the European Mediterranean region [134], chemical degradation resulting from heavy metals in China [137], waterlogging and soil salinity in India [138], or pesticides [139], physical degradation resulting from soil compaction [140], or environmental degradation due to invasive alien plants [141]. Several studies have been reported about different strategies for remediation of pollutants or other causes of soil degradation, which differ using plants or mushrooms. In the case of plants, remediation strategies depend on the type of pollutants (organic or inorganic) and the concentration. These strategies include bio-electrokinetic remediation [142], physicochemical remediation methods (such as chelate-assisted phytoextraction), phytohormones or plant growth regulators (e.g., auxins, abscisic acid, brassinosteroids, cytokinins, ethylene, gibberellins, jasmonic acid, polyamines, and nitric oxide), microbe-assisted remediation, plant growth-promoting rhizobacteria, inorganic/organic amendments such as biochar, compost, manure for immobilization pollutants in soils, as well as genetic strategies [143].

It is well known that environmental pollution was resulted from many anthropogenic activities such as speedy urbanization, agricultural practices, and rapid industrialization [144]. Several pollutants are involved in these activities, including metalloids, heavy metals, agrochemicals, radionuclides, fly ash, and organic compounds [145,146]. Phytoremediation is considered an environmentally and economically favorable technique using green plants to detoxify pollutants from contaminated water and soil. Phytoremediation has several mechanisms to remove, degrade, or immobilize the pollutants, such as accumulation (by phytoextraction or rhizofiltration), degradation (by rhizo-degradation or phytodegradation), dissipation (phytovolatilization), and immobilization by hydraulic control and phytostabilization [144]. This mechanism depends upon the type of pollutant, and plant species, which may utilize one or more of these mechanisms [147]. Phytoremediation of metals in polluted soil could be enhanced by using soil earthworm and arbuscular mycorrhizae [148]. The main mechanisms that plants can be used in remediation process may include bioremediation, phytoremediation, as reported by many studies such as Elallem et al. [149] Bhat et al. [150], Oladoye et al. [151], Gavrilescu [152], and Wang et al. [148].

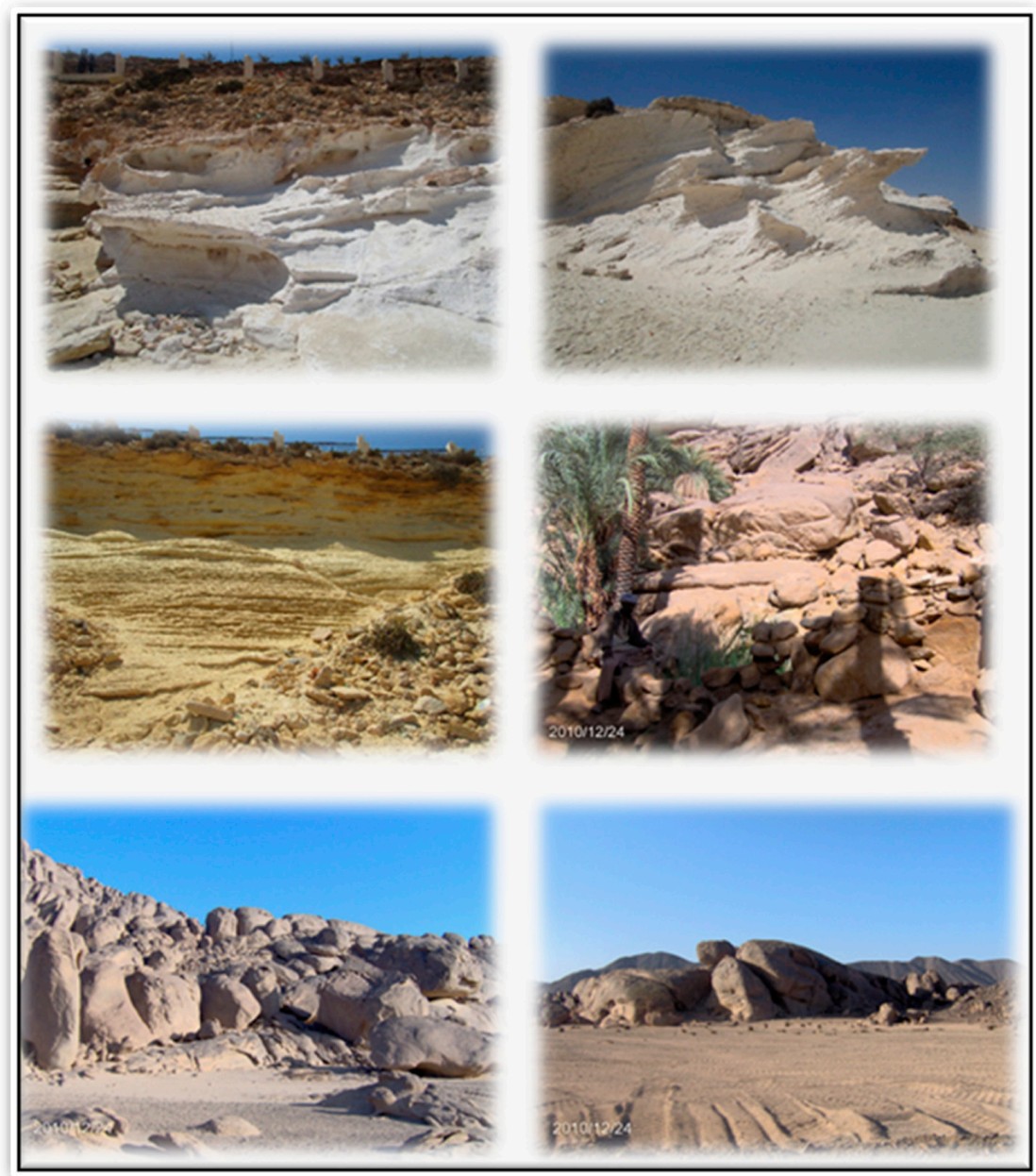

**Figure 7.** Soils that result from different geological forms definitely differ in their fertility and characterization, where sandy rocks form sandy soil (as a low-fertility soil, it needs much more treatment), whereas clay or loamy soil possesses better soil fertility. These photos show different geological forms from different places in Egypt (the first 3 photos from upper left from Matrouh; middle right photo from Shalateen; lower photos from Marsa Alam). All photos by **El-Ramady**.

**Soil degradation and its restoration mechanism**

**I. Soil degradation definition**

Soil degradation is defined as *"a change in the soil health status resulting in a diminished capacity of the ecosystem to provide goods and services for its beneficiaries. Degraded soils have a health status such, that they do not provide the normal goods and services of the particular soil in its ecosystem"* FAO (2022)

**II. Main soil degradation categories**

There are three main degradation categories under socio-economic and biophysical factors:
- Physical (soil sealing, compaction, erosion),
- Chemical (soil organic matter, pollution, salinization, and sodification), and
- Biological (soil fauna and flora and their biodiversity)

**Main soil restoration approaches**

**III. Soil restoration by plants**

Many degraded soil (polluted, marginal, submerged, eroded soil, etc.) need to restore by remove the reason of degradation by:
- Phytoremediation
- Bio-phytoremediation
- Nano- bio-phytoremediation
- Plant-soil feedback approaches

**IV. Soil restoration by mushrooms**

Mushrooms have the ability to restore of polluted soil and water through:
- Myco-remediation and bio-fermentation by enzymes (e.g., manganese peroxidase, laccase)
- Recycling of agro-wastes
- Producing biofertilizers, animal feeds, enzymes and bioethanol

**V. Mechanism of plant restoration**

Plant molecular, biochemical and physiological strategies could be used to restore soil by:
- Limitation of the absorption and transport,
- Chelation and sequestration of metals,
- Biosynthesis of signaling molecules (SA; NO)
- Induction of stress proteins, etc.

**VI. Mechanism of mushroom restoration**

Soil degradation can remediate by myco-remediation through many processes, which may include bio-degrade pollutants using mushrooms, through many mechanisms, such as biosorption, bioaccumulation, bioconversion, and biodegradation

**Figure 8.** This drawn figure is about soil degradation and its categories. It also includes a comparison between the role of higher plants and mushrooms in dealing with soil degradation through the process of soil restoration. Different mechanisms of the restoration process for both are highlighted. **Sources:** Bandyopadhyay [153], FAO [136], Ferreira et al. [134], Zhu et al. [154], and El-Ramady et al. [99].

Myco-remediation of polluted soils by mushrooms is an eco-efficient process of mushrooms to bio-degrade different pollutants through mechanisms such as bioaccumulation, biosorption, bioconversion, and biodegradation [99]. The main difference between remediation by plants and mushrooms is represented in the mechanism of this process, which mainly needs enzymes to be achieved in case of mushrooms, but in the case of plants, needs different approaches (Figure 10). The commonality between myco-remediation and phyto-remediation is that both are considered eco-friendly and capable of sustainable remediation of polluted soils as well as having many eco-benefits. The most important

benefits of myco-remediation are the possibility to produce biodiesel or bioenergy, enzymes and biofertilizers after the biodegradation process [36].

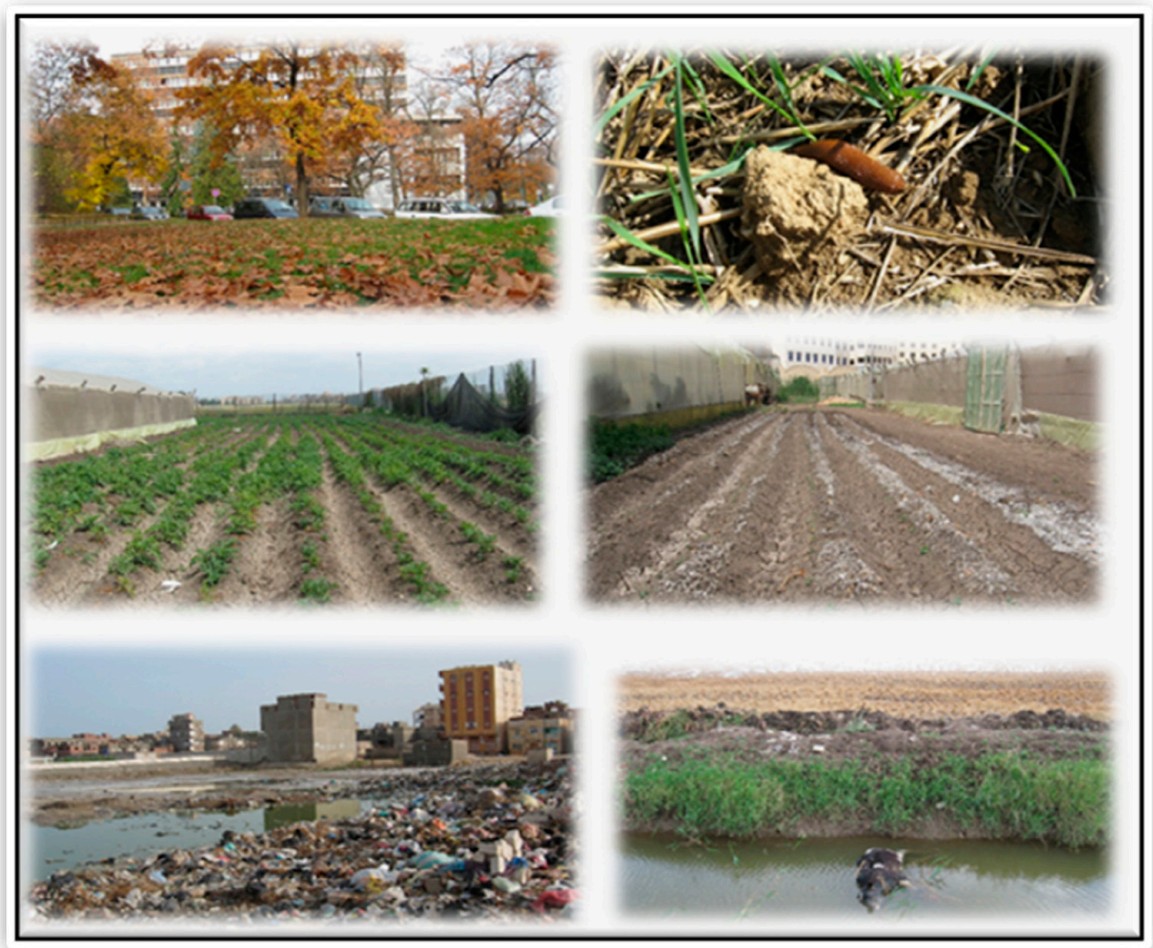

**Figure 9.** Soil/water pollution is considered an important global issue. These photos show the litterfall in the autumn season (**upper left photo**), which can enrich the soil with essential nutrients for plant growth and this reflects the biological activity in soil or soil fauna diversity in fertile soil (**upper right photo**). The middle photos represent soil salinity stress, which is considered a form of chemical soil degradation. The lower photos represent two reasons for soil degradation, including soil pollution and waterlogging (**lower left photo**) and soil pollution from dead animals in irrigation canals (**lower right photo**). The 2 upper photos are from Debrecen (Hungary), whereas the rest from Kafr El-Sheikh (Egypt). All photos by **El-Ramady**.

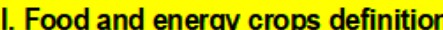

**Figure 10.** The integrated production of food and energy crops is illustrated in this drawn figure. The main differences between energy and food crops are listed, along with their limitations as well as different generations of biofuels with some examples. **Sources:** Saha et al. [155], and Goria et al. [156].

*7.1. Integrated Production of Food and Energy*

A great challenge faces the world to produce enough food for feeding the increasing population, especially under the limitations of global arable lands. This problem has a serious dimension if we need to answer this question: what is the main purpose of global land cultivation: biofuel production from energy crops or food production from food crops? This is a very difficult balance, which requires us to find and explore new lands, as-yet uncultivated, to cultivate and produce more food for the expected larger population in the future [157]. These lands are mainly problematic soils, which suffer from restrictions or obstacles preventing their productivity through the restoration processes such as polluted soils, marginal, waterlogged, submerged, degraded and salt-affected soils. The comparative items that are needed to identify the positive and negative points of each option may

include a large variety of economic, environmental, social, and sustainability aspects. Many suggestions could be applied as strategies and/or alternative choices for green energy supply, sustainability, and enough nutrition for the global growing population [157].

It is well known that the global population may reach 10 billion by 2050, which will lead to an increase in the global need for food, water, and energy (Figure 11). Along with energy, there is an urgent need to save sufficient quantities of clean water for different natural ecosystems and human societies [158]. Why is the integrated production of food and energy needed? Due to both food and energy being essential for human life and the limitation of land and water resources, the production of both should be integrated, where there is no life without all of them. It is logical to search for non-exploited lands for energy production while saving the arable lands for food production, especially under the amazing technologies in vertical expansion for both production systems to avoid the conflict of "food vs. fuel". Thus, many dimensions of the food–energy nexus have been explained in Table 4, which includes different factors that impact this nexus from different points of view. The importance of this nexus is increased when associated with water to be the food–energy–water nexus.

**Table 4.** Different forms of food–energy nexus and their impacts.

| Food-Energy Nexus | The Dimension of the Study | Refs. |
|---|---|---|
| Energy–food–water nexus | The performance of the energy–food–water nexus using solar energy under integrated production of fresh water from seawater desalination, biomass gasification and food systems in Qatar | [159] |
| Food–energy–water nexus | Reducing the losses in energy and water from consumer avoidable food wastes to increase sustainability in the food system in China | [160] |
| Food–energy nexus related to eco-pollution | Problems resulted from the production of energy and chemical fertilizers, as sources of environmental pollution due to the depletion of groundwater resources in Iran | [161] |
| Food–energy–water nexus | Identification of the change drivers in urban regions in China by a study of consumption of urban households, fixed capital formation and exports under food–energy–water system | [162] |
| Food–energy–land–water nexus | There is a need to produce a sustainable source of food, clean energy (biofuels), and water in Nigeria | [163] |
| Energy–food nexus | Collaborative management and conservation for scarcity of food and energy resources under climate policy were higher for low-income compared to high-income economies | [21] |
| Food–energy–water nexus | Mitigation of climate change and water circularity role in food–energy–water nexus for transition from a linear economy to a circular economy | [158] |
| Water–energy–nutrient–food nexus | Under urban agriculture system, water and nutrient needs at greenhouse farm and a container farm could be supplied by resources present in urban waters of wastewater and rainwater | [164] |
| Nutrient–food–energy–water nexus | Reusing urban wastewaters in urban farming can reduce energy needs for nutrient, water, irrigation, food transport, and wastewater pumping | [165] |

**Table 4.** *Cont.*

| Food-Energy Nexus | The Dimension of the Study | Refs. |
|---|---|---|
| Food–water–energy nexus | Integrated management in agricultural watershed and under drought can increase food production by 6% and reduce energy consumption by 3% compared to water-saving irrigation | [166] |
| Food–water–energy nexus | This nexus can contribute to sustainable and efficient management of different agricultural resources (i.e., energy, land, and water) | [167] |
| Food–energy–water nexus | This nexus governance depends on 9 principles: innovation, sharing, connectivity, participation, equitability, coordination, legitimacy, empowerment, and strategy | [168] |
| Energy–water–food nexus | Optimizing resilience can calculate to minimize emissions of $CO_2$ based on total profits, while considering natural disaster events as interruptions | [169] |
| Food–energy–water nexus | Recycling of food wastes as a source of energy and water can perform using a mechanical presser/anaerobic digester to produce biogas | [170] |
| Food–water–energy nexus | This nexus could examine the sustainability implications in China, which needs some strategies through developing socio-economic balance and saving resources from the consumption perspectives | [171] |

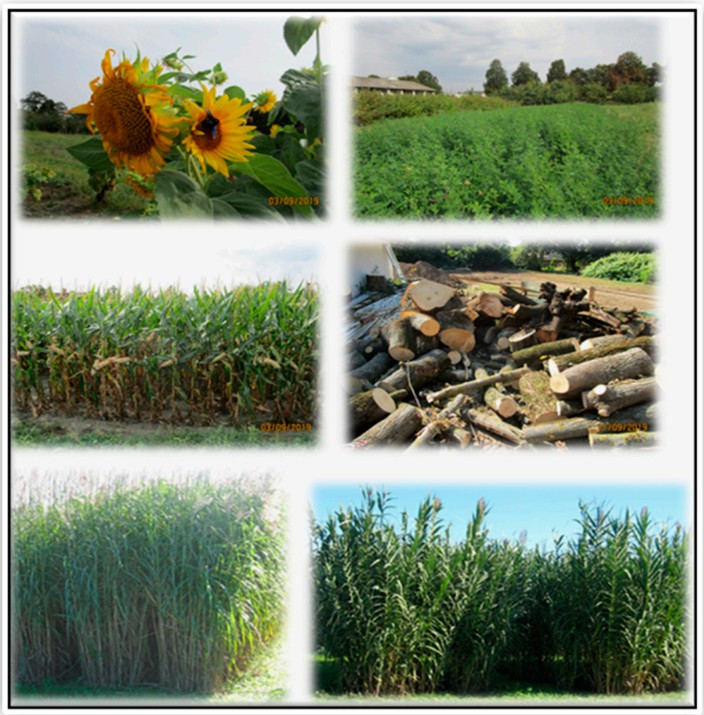

**Figure 11.** Several common energy crops, which can be used in energy generation for humans. It is also common to use wood as burning material for producing energy. These photos represent examples of energy crops; the upper-left photo is sunflower, to its right is alfalfa, the middle photos from the left, maize, in the right, wood materials. The lower-left photo is *Miscanthus* sp. And on the right is *Arundo donax* (as non-edible energy crops). All photos from the experimental farm, Debrecen University by **El-Ramady**.

Edible or food crops (e.g., potato, maize, sunflower, sugar cane, sugar beet, etc.) were the main candidates for the first generation of biofuels, whereas the residues of crops or forests or industrial wastes were presented the second generation seeking for clean energy and reduction of the dependence on fossil fuels [172]. Lignocellulosic biomass, as the most abundant bio-renewable materials in the world, can be produced from water and atmospheric $CO_2$ through photosynthesis using sunlight energy [173]. This biomass may represent forest residues, crop residues, industrial wastes, wood crops, municipal solid wastes, and food wastes [172]. The main herbaceous energy crops, which are considered the major sources of lignocellulosic biomass, include canary grass (*Phalaris canariensis*), switchgrass (*Panicum virgatum*), Miscanthus (*Miscanthus giganteus*), giant reed (*Arundo donax*), alfalfa (*Medicago sativa*), and Napier grass (*Pennisetum purpureum*) (Table 5). There is no subject discussed and reports on energy as published during 2022 (till 11 May 2022) by ScienceDirect about "*Crop and Energy*" 13,833 published materials, and a similar number (13,957) by SpringerLink. The future of renewable fuel resources and biofuel crops is the main target of climate change for several nations, such as China [174]. As is well known, each energy crop has certain growth conditions starting from the growing seeds or tubers till flowering and getting the harvested plant to produce the bioenergy as shown in Figure 12. The growing stage definitely differs from the processing stage for energy production, which depends on the plant species.

**Table 5.** An overview of some common energy crops and mushroom species.

| Energy Crop | The Common Meaning or Species in the Category |
|---|---|
| **I. Energy crops** | |
| *Lignocellulosic biomass* **(LCB)** *categories* | |
| 1. Annual and perennial energy grasses | Canary grass, switchgrass, Miscanthus, giant reed, alfalfa, and Napier grass |
| 2. Woody biomass | Natural forest residues, forestry wastes (wood chips, and branches from dead trees), tree bark, wood shavings, and sawdust |
| 3. Non-woody biomass | Agricultural wastes in the field (crop stubble, grasses, paddy husks, straw) and agricultural processing wastes (animal paunch waste, sugarcane bagasse, palm oil waste, cotton gin trash, etc.). |
| *Lignocellulosic biomass from crop residues* | |
| Biorefinery of crop residues applications | Converting biomass into bio-based products (biofuels, bioenergy, pharmaceuticals, biopolymers, surfactants) under circular bioeconomy Lignocellulose-degrading enzymes from microorganisms and their biotechnological applications |
| Biofuels | Bioethanol, biodiesel, biohydrogen, and biobutanol |
| Bioenergy | Biochar, biogas, syngas, methane, etc. |
| *Recent fractionation process of lignocellulosic biomass* | |
| | Pyrolysis, microwave assisted deep eutectic solvents, aldehydes, organo-Cat, hydrothermal and delignification |
| *Mechanism of biofuel production by plants* | |
| | Novel biofuels have been produced from LCB, such as bio-hydrogen, biobutanol, dimethylfuran by enzymes of cellulases, hemicellulases, lytic polysaccharide monooxygenase, ligninase, and cellobiose dehydrogenases |
| *Main food crops and their generated residues* | |
| | Apple (apple pomace), cotton (cotton sheets, cotton stalks), rice (rice straw, rice hulls), coffee (coffee husks, coffee pulp, wastewater), sugarcane (sugarcane bagasse, cane straw), barley (barley straw), beans (peel beans), sorghum (sorghum straw), orange (orange peel, orange bagasse), maize (corncobs, corn straw), soybean (soybean hull), wheat (wheat straw), grapes (grape pomace) |

**Table 5.** *Cont.*

| Energy Crop | The Common Meaning or Species in the Category |
|---|---|
| *Main biorefinery applications of some major crop residues* | |
| Coffee residues | Production of levulinic acid, gibberellic acid, biogas, bioethanol, biodiesel, α-amylase, pectinase, endoglucanase, and cultivation of mushrooms |
| Maize residues | Production of methane, prebiotic xylo-oligosaccharides, biosorbent, biogas |
| Soybean residues | Production of protease, β-amylase, α-amylase, biodiesel, biogas, bioethanol |
| Sugarcane residues | Production of glycosyl hydrolases, xylanases, endoglucanase, biobutanol, bioethanol, lignin, and levulinic acid |
| Rice residues | Production of cellulase, lignin degrading enzymes, biochar, nano-silica, nanocrystals, biobutanol, lignin, and cellulose |
| Wheat residues | Production of bioethanol, biogas, levulinic acid, bacterial cellulose, and mushroom cultivation |
| **II. Mushrooms** | |
| *Most important mushroom species produce bioethanol* | |
| | *Pleurotus florida*, *P. ostreatus*, *Ganoderma lucidum*, *Lentinula edodes* |
| *Mechanism of biofuel production by mushroom* | |
| | Production of biofuels and energy from LCB is based on biochemical processes, which LCB needs C:N ratio < 30 and humidity >30% through degrading enzymes (laccase, mannanase, cellulase, xylanase, etc.) |
| *Spent mushroom substrates* (**SMS**) *and its use for bioethanol production* | |
| | I. SMS of both mushrooms (*Agaricus bisporus* and *Pleurotus forida*) used for lignocellulolytic enzymes (hydrolytic and oxidative enzymes) II. SMS of *Lentinula edodes* was used for enzymatic saccharification, which resulted in high glucan digestibility (80–90%) in the SMS beside phenolics III. Using Hot-air (75–100 °C) pasteurization instead of autoclaving for SMS of *Lentinula edodes* by enzymatic digestibility of glucan in SMS IV. SMS of *Ganoderma lucidum* used by 0.2% (*v/v*) for fermentation using baker's yeast (*Saccharomyces cerevisiae*) and incubated for 5 days at 30 °C |
| *Fermentation using mushroom for bioethanol production* | |
| | I. Fermentation using *Pleurotus florida* on cotton spinning wastes and the optimum ethanol yield (1.18 g $L^{-1}$) was obtained by 64% at 60 h II. Mushroom of *Dictyopanus genera* can its enzyme (laccase activity 267 U $L^{-1}$) from oil palm delignification process for bioethanol derived-cellulose |

Sources: [99,172,173,175–182].

### 7.2. Integrating Food Crops and Mushrooms

Integration in agricultural sectors is very important as a general target for sustainable agriculture, in particular the production of food and energy. Mushrooms represent an important source of non-traditional food for human nutrition. So, the marketing of mushrooms includes cultivated edible (54%), medicinal mushrooms (38%), and wild mushrooms (8%) [183]. Mushrooms are also being considered as second-generation biofuels, and this does not compete with the production of foods; the cultivation of mushrooms and their waste recycling should be exploited for a circular bioeconomy [184]. Agro-wastes could be also bio-converted and composted with wastes of dairy foods during the cultivation of mushrooms [185]. Cultivation of edible and/or medicinal mushrooms is a sustainable food–energy production system because they can mainly cultivate agro-wastes and/or agro-industrial residues as well as the wastes of their cultivation (spent mushroom substrate or SMS) themselves, which can be used in bioenergy production. Therefore, this combination of food–energy production by mushrooms is considered an integrated form of production as reported in the case of production of mushrooms and tomatoes under the circular bioeconomy approach [186]. The co-cultivation of plants (e.g. vegetables) and

mushrooms has been reported in many studies, which leads to higher yields and higher biomass production, due to the impacts of mushroom hyphae in increasing the availability of water and minerals around plant roots [186,187].

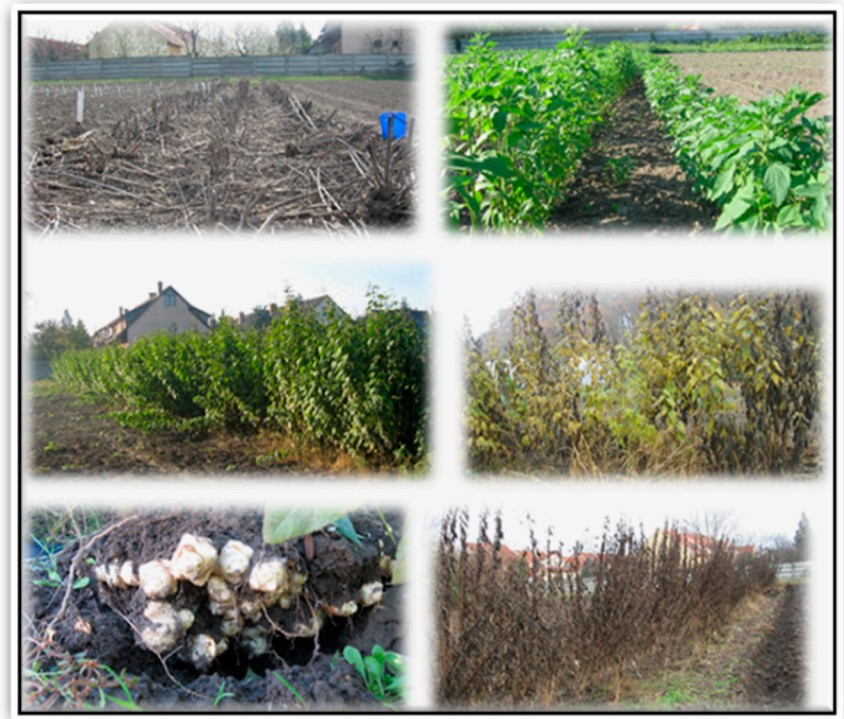

**Figure 12.** Jerusalem artichoke as an example of energy crop and as a biorefinery crop with focus on different growth stages. These photos start from the upper left, which presents the storing of some tubers in the soil from the previous crop until March, and then the tubers were collected from soil and put in containers in refrigerator until the cultivation in April, where the right photo after cultivation shows the tubers in June. The middle photos represent the plants during October and November. The lower photos: left is the tubers in the mature stage during December and the right photo for plants during January in winter. All photos were taken from the experimental farm, Debrecen University by **El-Ramady**.

The integrated relationship between mushrooms and higher plants does not only include co-cultivation at the same time, but also include many benefits in different applications, such as:

(1)  Improving mushroom (*Pleurotus ostreatus*) production cultivated on staple crop residues including banana, cassava, common bean, and maize [188],

(2)  Utilizing fruit waste substrates (peels of avocado, orange, and pineapple) in mushroom (*Pleurotus eryngii* and *P. ostreatus*) production [189],

(3)  Using hulls of faba bean as substrate for mushroom (*Pleurotus ostreatus*) cultivation and for animal feed production [190],

(4)  Producing spawns from banana leaf-midribs for cultivation of oyster (*Pleurotus ostreatus*) mushrooms [191],

(5)  Integrating mushroom cultivation and production in a circular agro-system into food chains [192], and

(6)  Using spent mushroom compost of mushroom (*Agaricus subrufescens* and *A. bisporus*) for cultivation of lettuce, tomato, and/or cucumber in a sustainable system in the same container under greenhouse conditions [193].

There are many species of mushrooms that have the ability to produce bioethanol, such as *Pleurotus florida*, *P. ostreatus*, *Ganoderma lucidum*, *Lentinula edodes*, etc., as reported by Xiong et al. [178], Rueda et al. [182], Sudhakar et al. [179], Devi et al. [177],

Periyasamy et al. [181], and Ranjithkumar et al. [180] (Table 5; Figure 13). The suggested mechanism of producing biofuels and energy from lignocellulosic biomass (LCB) is based on biochemical processes, which need certain conditions, especially C:N ratio < 30 and humidity >30% through biodegrading enzymes (e.g., laccase, mannanase, cellulase, and xylanase).

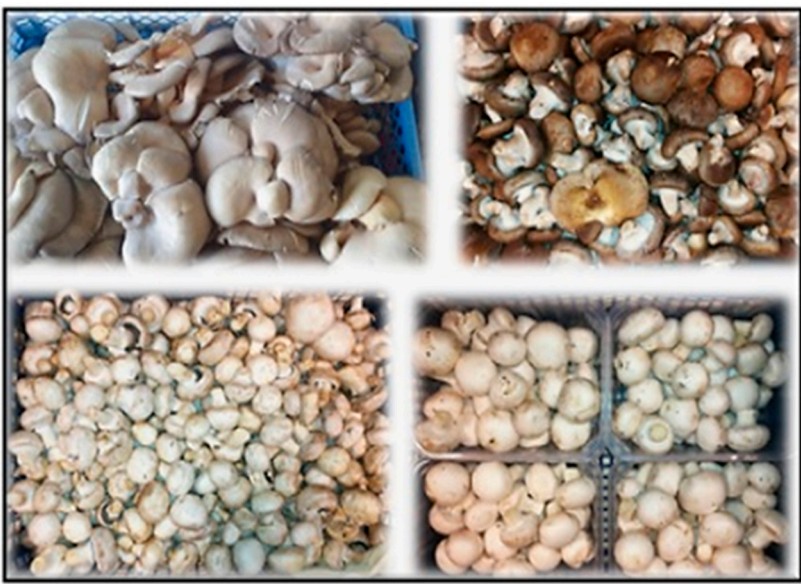

**Figure 13.** Some common fresh edible mushrooms from the upper photos left (*Pleurotus ostreatus*) to right (*Lentinula edodes*) and in the lower photos: white champignon mushroom (*Agaricus bisporus*) left (quality: II. class) and right (quality: I. class). All photos by **Gréta Törős** (Debrecen Uni.) from Magyar Gomba Kertész Kft.

## 8. General Discussion

A variety of applications of mushrooms in the agricultural, medicinal, and pharmaceutical fields are well known. Although this review focuses on the comparison between mushrooms and higher plants, the findings will have a bearing on the co-cultivation of mushrooms and cultivated plants as an agro-sustainable approach. Returning to the main questions posed at the beginning of this study, it is now possible to state that the answer to the previous questions could be mentioned in brief as follows:

What is the preferable for soil restoration food crops or energy crops, or mushrooms? In general, energy crops (especially non-edible crops) are preferable plants for problematic soils, since they have the ability to reclaim these soils, are tolerant to stressful conditions in these soils, and could save arable soils for food production. Mushrooms have many benefits in the respiration of degraded soils through many biochemical processes resulting from the enzymatic activities of mushrooms, such as decomposition, biodegradation, bio-weathering, bioconversion, and biosorption. Mushrooms also have the ability to increase soil aggregates, promote biodegradation of pollutants, and enhance the nutrient bioavailability in soil [194].

What is the possibility for sustainable production of both food crops and mushrooms? The sustainable agriculture for both mushrooms and cultivated plants could be achieved under the co-cultivation of mushrooms and vegetables as an integrated system for foods and mushrooms production. This approach will prove useful in expanding our understanding of how the cultivated mushrooms can support the growth and productivity of co-cultivated vegetables at the same time. Co-cultivation of mushrooms and lettuce can reduce the accumulated $CO_2$ emission into the air by 80.6%, due to the ability of mushrooms to support cultivated lettuce by $CO_2$ resulting from the respiration of mushrooms. This continuous cultivation system of both mushrooms and lettuce can reduce $CO_2$ emissions into the air and achieve sustainable agriculture [195]. The circular bioeconomy approach could be realized for the integrated production of foods and mushrooms. Different agro-

industrial residues could be managed in a sustainable approach through motivating the use of these residues for the production of both vegetables and mushrooms or fungi culture.

## 9. Conclusions and Future Perspectives

This is the first study to report a photographic comparison between mushrooms and plants. This comparison that we have identified therefore assists in our understanding of the role of both mushrooms and cultivated plants especially towards the circular biorefinery approach. The findings from this study make several contributions to the current literature. First, several similar and differences are be noticed between mushrooms and plants. Second, the most important difference lies in the fact that plants have the ability to make their own food (autotrophic), due to the existence of chlorophyll, whereas mushrooms are saprophytic organisms. Third, the co-cultivation of mushrooms and vegetables is an integrated and sustainable production system. Fourth, mushrooms are considered a sustainable solution for bioenergy and biorefinery. Although the study has successfully demonstrated that mushrooms and plants could be integrated in many different agricultural, medicinal, and pharmaceutical practices, it has certain limitations in terms of the harmony between mushrooms and cultivated crops. It is suggested that the association of these factors is investigated in future studies to make precise criteria for this co-cultivation of mushrooms and plants. It is suggested that before the generalization is introduced, many co-attributes between mushroom and plants should be carried out on the phytomedicinal and ecotoxicological attributes. This review also focused on the phytomedicine and its potential for human health, the unconventional foods derived from plants and mushrooms, soil degradation and its restoration by plants and mushrooms, the integrated production of food and energy, and finally the agro-integration between food crops and mushrooms. This review may open new windows concerning the urgent strategy for producing food and energy at the same time.

**Author Contributions:** J.P. and H.E.-R. developed the idea and outline of the review. The 2nd and 3rd sections were written by K.B., the 4th and 5th sections by N.A. and the 6th section by M.E.E.-M. The 7th section was written by H.E.-R., whereas the 8th and 9th sections were written by G.T., X.L. and P.H. Both H.E.-R. and K.B. revised the manuscript thoroughly and finalized it. All authors have read and agreed to the published version of the manuscript.

**Funding:** The research was supported by the Stipendium Hungaricum Scholarship Program (SH ID:140993). The work/publication is supported by the EFOP-3.6.3-VEKOP-16-2017-00008 project. The project is co-financed by the European Union and the European Social Fund.

**Institutional Review Board Statement:** Not applicable.

**Informed Consent Statement:** Not applicable.

**Data Availability Statement:** Not applicable.

**Acknowledgments:** The authors are grateful for the support of the 2020-1.1.2-PIACI-KFI-2020-00100 Project "Development of innovative food raw materials based on Maillard reaction by functional transformation of traditional and exotic mushrooms for food and medicinal purposes".

**Conflicts of Interest:** The authors declare no conflict of interest.

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
