# Peer review of "A Comparative Photographic Review on Higher Plants and Macro-Fungi: A Soil Restoration for Sustainable Production of Food and Energy"

_sustainability, doi:10.3390/su14127104_

Round 1
Reviewer 1 Report
In my opinion, the manuscript (A Comparative Photographic Review on Higher Plants and Macro-Fungi: A Soil Restoration for Sustainable Production of Food and Energy) is interesting. It would give insights into mushrooms and plants taken into account for food and energy. The idea of the manuscript is most demandable and reasonable for managing sustainable food security. Climate change tigress burden on soil fertility as well as crop production at the global level, and this article may be useful for scholars who are working in the concern field. The language of the must be improve in revised manuscript and add suitable relevant references; it can be improved further with the following suggestions-
Phytoremediation section must be restructure and add few examples of emerging pollutant removal studies.
Line 365-367 please elaborate the methods of phyto-extraction, phyto-volatilization, phyto-immobilization, phyto-stabilization etc. See references https://doi.org/10.1002/ldr.4090 and https://doi.org/10.3390/nano12050769.
Hypothesis, objective much correlate with conclusion, please re-frame the sentences.
Author Response
The authors would like to thanks for Reviewer comments that improve our manuscript quality.
Comments and Suggestions for Authors
In my opinion, the manuscript (A Comparative Photographic Review on Higher Plants and Macro-Fungi: A Soil Restoration for Sustainable Production of Food and Energy) is interesting. It would give insights into mushrooms and plants taken into account for food and energy. The idea of the manuscript is most demandable and reasonable for managing sustainable food security. Climate change tigress burden on soil fertility as well as crop production at the global level, and this article may be useful for scholars who are working in the concern field. The language of the must be improved in revised manuscript and add suitable relevant references; it can be improved further with the following suggestions-
Response: Many thanks for your great words and your support!
The MS was edited and some relevant refs. Were added to the revised MS, thanks again!!
Phytoremediation section must be restructure and add few examples of emerging pollutant removal studies.
Line 365-367 please elaborate the methods of phyto-extraction, phyto-volatilization, phyto-immobilization, phyto-stabilization etc.
See references https://doi.org/10.1002/ldr.4090
and https://doi.org/10.3390/nano12050769.
Response: Section was added including suggested refs. In the revised MS from line 368 to 384.
Hypothesis, objective much correlate with conclusion, please re-frame the sentences.
Response: Added in the revised MS from line 579 to 583.

Reviewer 2 Report
1. To make it clear to readers, I suggest adding some concise descriptions of how the paper is organized at the end of section 1.
2. The authors demonstrated that Chinese herbal medicines are very common and widely used for treating several diseases, and how medicinal plants were considered in this comparative study? Please clarify it.
3. Some figures are out obvious, they should be obvious.
4. The list of references is too long.
Author Response
The authors would like to thanks for Reviewer comments that improve our manuscript quality.
Comments and Suggestions for Authors
- To make it clear to readers, I suggest adding some concise descriptions of how the paper is organized at the end of section 1.
Response: Many thanks for your great suggestion if you mean to add the Table of contents!
If not, in the section of 2. Methodology of the Review? All details about how this MS was organized, thanks!
Table of contents
- Introduction
- Methodology of the Review
- Plant and human nutrition for sustainability
- Phytomedicine and human health
- Higher plants and mushrooms: a general comparison
- Unconventional foods of plants and mushrooms
- Soil restoration by plants and mushrooms
7.1 Integrated production of food and energy
7.2 Integrating food crops and mushrooms
- General discussion
- Conclusions and future perspectives
References
- The authors demonstrated that Chinese herbal medicines are very common and widely used for treating several diseases, and how medicinal plants were considered in this comparative study? Please clarify it.
Response: Many thanks for your comment! More details were added to revised MS in line 220 to 224!
- Some figures are out obvious, they should be obvious.
Response: Many thanks for your comment! All figures were added and changed their captures! For more explanations to the readers!
- The list of references is too long.
Response: Please let me say I can not please, because the other 2 reviewers asked for more adding refs. Please accept my apology, thanks!

Reviewer 3 Report
Manuscript ID: sustainability-1749954
Title: A Comparative Photographic Review on Higher Plants and Macro-Fungi: A Soil Restoration for Sustainable Production of Food and Energy
The manuscript is unquestionably publishable. The contribution of the authors' work in preparing this very good review work should be appreciated. The authors, in this comprehensive review, provided comprehensive information on the understanding of the role of both mushrooms and cultivated plants. The manuscript has been carefully prepared, both from a scientific and editorial point of view. The work does not contain any large errors, only a few small editorial errors contained in the attached PDF file.
Kind regardsReviewer

Author Response
The authors would like to thanks for Reviewer comments that improve our manuscript quality.
Comments and Suggestions for Authors
Manuscript ID: sustainability-1749954
Title: A Comparative Photographic Review on Higher Plants and Macro-Fungi: A Soil Restoration for Sustainable Production of Food and Energy
The manuscript is unquestionably publishable. The contribution of the authors' work in preparing this very good review work should be appreciated. The authors, in this comprehensive review, provided comprehensive information on the understanding of the role of both mushrooms and cultivated plants. The manuscript has been carefully prepared, both from a scientific and editorial point of view. The work does not contain any large errors, only a few small editorial errors contained in the attached PDF file.
Kind regards
Reviewer
Response: Many thanks for your great words and your support!
All comments were done, thanks!
